# Harnessing synthetic active particles for physical reservoir computing

Xiangzun Wang[1,2] & Frank Cichos [1]✉

The processing of information is an indispensable property of living systems realized by networks of active processes with enormous complexity. They have inspired many variants of modern machine learning, one of them being reservoir computing, in which stimulating a network of nodes with fading memory enables computations and complex predictions. Reservoirs are implemented on computer hardware, but also on unconventional physical substrates such as mechanical oscillators, spins, or bacteria often summarized as physical reservoir computing. Here we demonstrate physical reservoir computing with a synthetic active microparticle system that self-organizes from an active and passive component into inherently noisy nonlinear dynamical units. The self-organization and dynamical response of the unit are the results of a delayed propulsion of the microswimmer to a passive target. A reservoir of such units with a self-coupling via the delayed response can perform predictive tasks despite the strong noise resulting from the Brownian motion of the microswimmers. To achieve efficient noise suppression, we introduce a special architecture that uses historical reservoir states for output. Our results pave the way for the study of information processing in synthetic self-organized active particle systems.

Storing and processing of information is vital for living systems[1]. The detection of low amounts of chemicals by a bacterium to navigate environments[2,3], the feedback mechanisms controlling and maintaining the function of organisms[4], or the highly sophisticated computations in large biological neural networks in the brain[5] are intricate examples of this importance created by evolutionary development. All these processes with living matter as the substrate of computation rely on its inherent activity, e.g., the microscopic energy conversion to power the signaling cascades in the presence of strong thermal noise. They have inspired many computational models of machine learning that are not executed on living matter, but on well-designed electronic hardware using completely different information representations than living matter[6]. Recurrent neural networks are a variant of such mathematical algorithms with a fading memory that allows learning from information sequences as in language or time series[7]. Reservoir computers employ sparsely and statically connected recurrent nodes[8–10] or even a single node by using time-multiplexing[11,12] to create a high-dimensional space. Information can be injected into this space to spread over the many degrees of freedom. Unlike the training of other neural networks, where the interactions of all components are optimized, training reservoir computers is often only restricted to finding how the desired information can be retrieved from the node states using adjustable readout only[13,14].

As one of the main properties of recurrent nodes is the memory of past states, reservoir computers also allow for a physical realization on unconventional computational substrates[15–17] using optoelectronic oscillators[18–20], mechanical oscillators[21,22], carbon nanotubes[23] or passive soft bodies[15] as excitable physical systems. It is thus intriguing to close the loop and draw inspiration from active living systems to explore microscopic reservoir computing in synthetic active microsystems, where noise is omnipresent as well but a precise control over the shape and the physics of the active system is possible. Motile synthetic active particles have generated enormous interests as a model for self-propelled systems far from equilibrium and emergent

[1]Peter Debye Institute for Soft Matter Physics, Leipzig University, 04103 Leipzig, Germany. [2]Center for Scalable Data Analytics and Artificial Intelligence (ScaDS.AI) Dresden/Leipzig, 04105 Leipzig, Germany. ✉e-mail: cichos@physik.uni-leipzig.de

collective effects[24] that mimic, for example, the dynamics of swarms[25–27]. Information processing[28,29] and machine learning[30] in experiment[31] and simulation[32–36] have also entered the field of synthetic active matter. However, studies that extend the use of synthetic microparticles as computational substrates are still rare or purely numerical[37].

We demonstrate that motile self-propelled active microparticles can be used for physical reservoir computing. An active particle self-organizes into a nonlinear dynamical unit based on a retarded propulsion towards an immobile target forming a noisy physical recurrent node. The node is perturbed by a time-multiplexed input signal to form a network of virtual nodes with sparsely connected topology. Multiple of these active units realize a high dimensional space of our reservoir computer. Harnessing the physics and inherent dynamics of the active particles, the reservoir computer is capable of predicting chaotic series despite the strong influence of the Brownian motion of the active particles. In particular, we find that using historical reservoir states for the output derivation effectively suppresses the intrinsic noise of the reservoir opening new routes for reservoir computing in noisy systems.

## Results

### Active particle recurrent node

A reservoir computer (RC), as a paradigm derived from recurrent neural networks, consists of recurrent nodes that nonlinearly process external signal inputs as well as their previous states[8,10]. We realize a simple recurrent node with the help of a single synthetic active particle as a microscopic model for motile active matter[24]. An active particle is a polymer microbead of 2.19 μm diameter with the surface decorated by gold nanoparticles (about 8 nm diameter). It is immersed in a thin layer of water bounded by two coverslips and can move freely in two

dimensions. The active particle is propelled with a speed $v_0$[38] towards an immobile target particle by partially heating the gold nanoparticles using a focused laser in a microscopy setup (see Fig. 1a, b and Methods section). The continuous propulsion in a desired direction is realized by adjusting the focused laser spot position in real time with the help of a feedback loop using a spatial light modulator (SLM). A time delay $\delta t$ is added to the intrinsic feedback latency to control the active particle[27,39] and to introduce a finite reaction time, as is inherent in many processes in living systems. Correspondingly, the attraction $\hat{\mathbf{F}}(t)$ experienced by the active particle at time $t$ is determined by its position at previous time $t - \delta t$,

$$\hat{\mathbf{F}}(t) = -\frac{\mathbf{r}(t - \delta t)}{|\mathbf{r}(t - \delta t)|}, \tag{1}$$

where $\mathbf{r}$ denotes the location of the active particle with respect to the immobile particle center. The consequence of this retardation is an angular displacement $\theta(t) = \phi(t) - \phi(t - \delta t)$ during the delay time with $\phi$ denoting its angular position, thus results in a transient rotational motion of the active particle around the immobile target (Fig. 1c). The dynamics of the angular position can then be described by a nonlinear delay differential equation

$$\dot{\phi}(t) = \frac{v_0}{R_0} \sin(\theta(t) + u(t)) + \frac{\sqrt{2D}}{R_0} w(t) \tag{2}$$

assuming that the active and the immobile particle (radius of $a_{act}$ and $a_{imm}$) are in physical contact, i.e., $R_0 = a_{act} + a_{imm}$, under the delayed attraction. The active particle in water is subject to Brownian motion as represented by the noise term in Eq. (2) with $w(t)$ denoting Gaussian white noise. The diffusion coefficient was determined from the

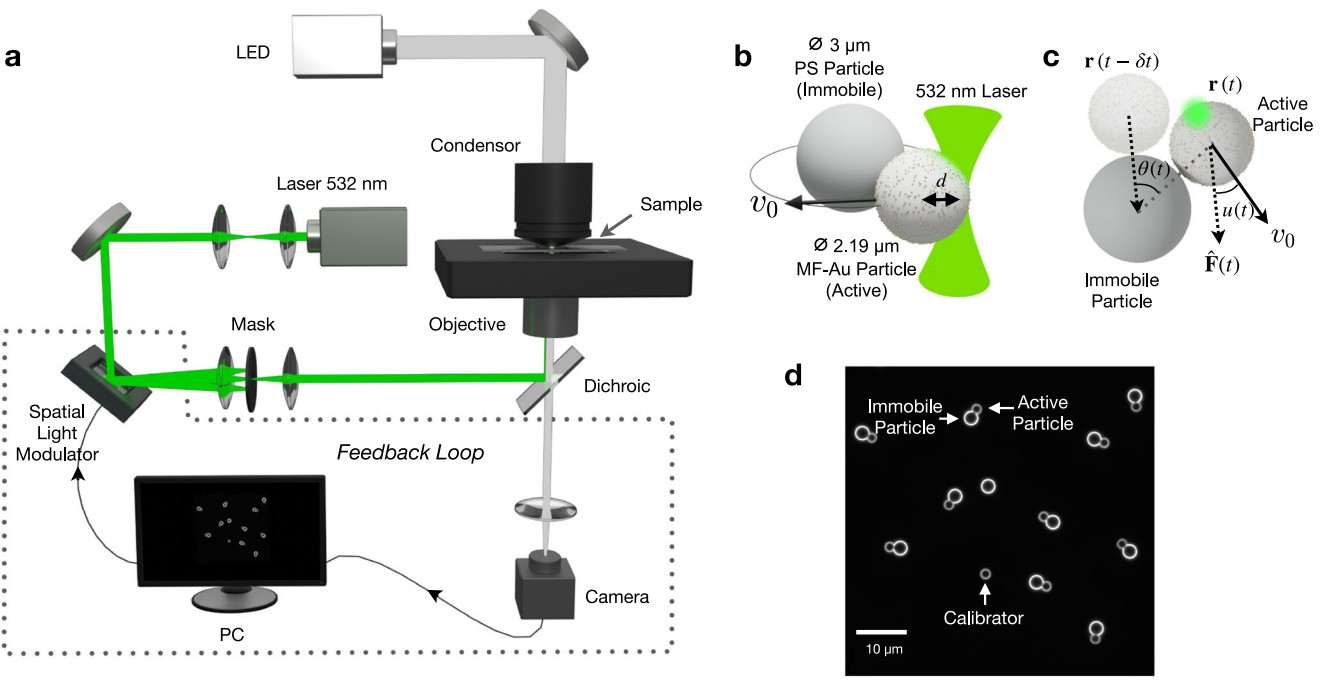

**Fig. 1 | Experimental realization. a** Experimental setup (see Sec. Method for detail). **b** Active particle recurrent unit consisting of a gold-nanoparticle covered melamine resin particle (MF–Au) and an immobile target polystyrene particle (PS). A 532 nm laser is focused on the active MF–Au particle at a distance $d$ from the particle center. The resulting heat and asymmetric temperature induce a self-thermophoretic motion of the particle with a speed of $v_0$ and a direction set by the vector from the laser to the particle center. **c** Top view of the active particle system. The active particle is controlled to carry out a motion along $\hat{\mathbf{F}}$ towards the immobile

particle with a time delay $\delta t$. The direction of $\hat{\mathbf{F}}(t)$ (dashed arrow) is determined by the previous active particle location $\mathbf{r}(t - \delta t)$. An additional angle $u(t)$ between the particle propulsion direction (solid arrow) and $\hat{\mathbf{F}}(t)$ represents an external input into the system. **d** Darkfield microscopy image of the sample consisting of 10 active-immobile particle pairs (larger circle is the immobile particle, smaller circle the active particle) as physical nodes in the experiment. An additional calibrator is used as an active particle swimming along a square route to measure the propulsion speed $v_0$. The real-time video is provided in Supplementary Movie 1.

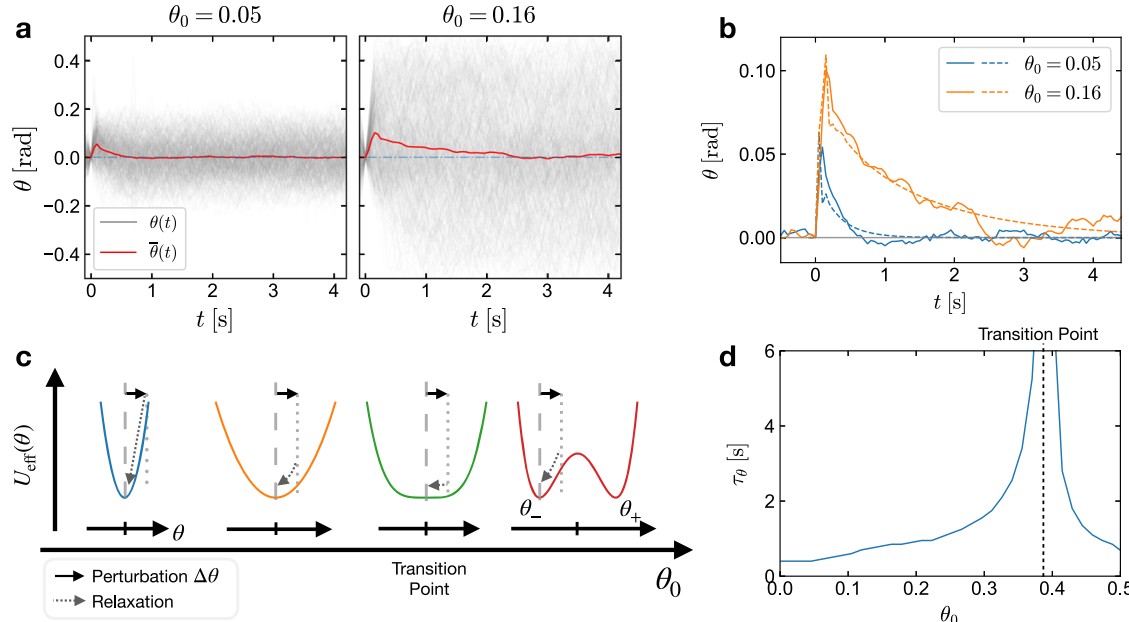

**Fig. 2 | Impulse response of the active particle node. a** Responses of $\theta(t)$ to an impulsive input $u(t) = \delta(t)$ measured in the experiment with $\theta_0$ of 0.05 and 0.16. The gray curves denote the measured $\theta(t)$ traces, which strongly fluctuate due to the Brownian motion of the active particles. The red curves denote the means of 500 $\theta(t)$ traces in each case. **b** Comparison of the mean impulse responses of $\theta(t)$ from the experiment (solid lines), same as $\bar{\theta}(t)$ in (**a**) to the ones obtained from the deterministic simulation (dashed lines). **c** Effective potential $U_{eff}(\theta)$ with different $\theta_0$. $U_{eff}(\theta)$ transitions from a single well to a double well form at the transition point, with its local minima positions bifurcating from zero to two opposite values ($\theta_+, \theta_-$). After perturbations $\Delta\theta$ (solid arrows), $\theta$ relaxes to one of the minima (dashed arrows). **d** Relaxation time $\tau_\theta$ of $\theta$ as function of $\theta_0$ evaluated in deterministic simulations. $\theta$ is perturbed with $\Delta\theta$ from one of the states $\theta_{+,-}$ at $t = 0$ s, then relaxes to $\theta(\tau_\theta) - \theta_{+,-} = \Delta\theta/10$. $\tau_\theta$ diverges to infinity at the transition point of $U_{eff}(\theta)$ (dotted line).

experiment to be $D = 0.08\ \mu m^2 s^{-1}$, giving rise to a Péclet number of $Pe = a_{act}v_0/D = 38.7$. To make the physical recurrent node capable of receiving external inputs, we introduce the $u(t)$ in Eq. (2) representing an angular deviation of the particle propulsion direction from $\hat{\mathbf{F}}$ (Fig. 1c).

For its function as a recurrent node, the dynamics of the angle $\theta(t)$ is important. Described by Eq. (2), the dynamics of $\theta$ can be approximated by an overdamped motion of a particle in a self-generated effective quartic potential $U_{eff}(\theta)$[27], which resembles a generic Landau-type description[27,40]. The shape of the potential is controlled by a dimensionless parameter $\theta_0 = v_0\delta t/R_0$. With increasing $\theta_0$, $U_{eff}(\theta)$ transitions from a near parabolic shape with a minimum at $\theta = 0$ to a symmetric double well shape with minima at $\theta_+$ and $\theta_-$ in a pitchfork bifurcation (see Fig. 2c and Supplementary Note 1).

A perturbation of $\theta$ (solid arrow in Fig. 2c) will result in a relaxation with a dynamics determined by the control parameter $\theta_0$. We have experimentally determined the response of $\theta(t)$ to an impulsive perturbation $u(t) = \delta(t)$ for different values of $\theta_0$ (Fig. 2a, b) active particle. However, the ensemble average of 500 trajectories of $\theta(t)$ (red lines in Fig. 2a) exhibits an asymptotic behavior, which nicely reflects the response evaluated in the deterministic simulation (Fig. 2b dashed lines). The characteristic relaxation time $\tau_\theta$ extracted from the deterministic simulations reveals the expected strong increase around the transition point. Tuning the control parameter $\theta_0$, i.e., activity ($v_0$) and/or delay ($\delta t$) therefore allows to manipulate the fading memory of the active particle as a recurrent node, which is paramount for the coupling and response of the recurrent nodes in our reservoir computer. Note that the nonlinear dynamics of the active particle system is the result of the delayed propulsion towards the target.

**Reservoir computer with active particle nodes**
The asymptotic relaxation of $\theta(t)$ demonstrated in Fig. 2**b** represents a basic requirement for reservoir computing[9,41–43]. The time delay also realizes the coupling of the recurrent nodes with their past states. In the discrete-time setting of our experiment with the sampling period $\Delta t$, Eq. (2) can be rewritten as

$$\phi(T) = \phi(T-1) + \frac{v_0\Delta t}{R_0}\sin(\theta(T-1) + u(T-1)) + \frac{\sqrt{2D\Delta t}}{R_0}W(T-1),$$

(3)

where $T = t/\Delta t$ is an integer number representing the time step. $W$ denotes Gaussian random numbers with zero mean and unit variance. The evolution of $\theta$ then follows as

$$\begin{aligned}
\theta(T) &= \phi(T) - \phi(T - \delta T)\\
&= \theta(T-1) + \frac{v_0\Delta t}{R_0}\sin(\theta(T-1) + u(T-1))\\
&\quad - \frac{v_0\Delta t}{R_0}\sin(\theta(T - \delta T - 1) + u(T - \delta T - 1))\\
&\quad + \frac{\sqrt{2D\Delta t}}{R_0}(W(T-1) - W(T - \delta T - 1)),
\end{aligned}$$

(4)

with the discrete time delay $\delta T = \delta t/\Delta t$. Referring to the concept of virtual nodes and time-multiplexing[12], we consider the transient state $\theta(T)$ of a physical node at different time steps as virtual nodes constituting the reservoir. Each virtual node state $\theta(T)$ is, according to Eq. (4), coupled to its previous states $\theta(T-1)$ and $\theta(T - \delta T - 1)$. The virtual nodes thus reflect a topology with sparse interconnections realized by the delay of the physical node (Fig. 3a). The interconnections are inherently nonlinear due to the *sine* function in Eq. (4), which originates from the physical interaction between the active and the immobile particle and naturally serves as the activation function in our RC.

The working principle of our RC is now illustrated in Fig. 3. For simplicity of the discussion, we consider a RC with a single physical node, scalar input $X_n$ and output $Y_n$ at the $n$-th computation step (for the general case see Supplementary Note 2). The input layer $\mathbf{u}_n$ is

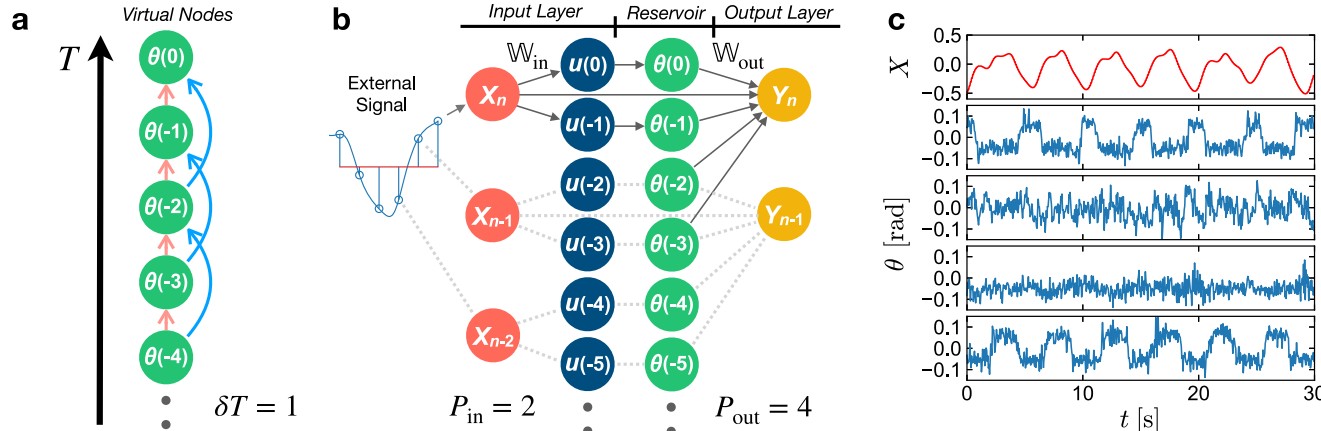

**Fig. 3 | Architecture of the reservoir computer. a** Topology of the reservoir with a single physical node and a discrete time delay $\delta T = 1$ in this example. The $\theta$ of the active particle at different time steps $T$ are considered as the virtual nodes. Each virtual node state $\theta(T)$ is nonlinearly coupled to previous states $\theta(T-1)$ and $\theta(T-\delta T-1)$ through the dynamics of the system Eq. (4). **b** Sketch of the information processing in the single node RC with $P_{in} = 2$ and $P_{out} = 4$ in this example. The external signal $X$ of each computation step is multiplexed by a weight matrix $\mathbb{W}_{in}$ to

generated via a matrix of input weights $\mathbb{W}_{in} \in \mathbb{R}^{P_{in} \times 2}$,

$$\mathbf{u}_n = \mathbb{W}_{in}[b_{in}, X_n]^T, \qquad (5)$$

with the scalar $b_{in}$ as the input bias. $\mathbf{u}_n$ is a one-dimensional array containing $P_{in}$ elements, which are sequentially input into the physical node as the angle $u(T)$ of the active particle in Eq. (3). This operation is equivalent to a time-multiplexing of the input $X_n$ into $P_{in}$ virtual nodes with $\mathbb{W}_{in}$ as the mask, as commonly applied in continuous-time single node physical RC approaches[12,44]. The output $Y_n$ is derived as a linear combination of a scalar bias $b_{out}$, the input signal $X_n$, and the node states $\theta$ of the past $P_{out}$ time steps using an output weight matrix $\mathbb{W}_{out} \in \mathbb{R}^{1 \times (2 + P_{out})}$,

$$Y_n = \mathbb{W}_{out}[b_{out}, X_n, \theta(T=0), \theta(T=-1), \cdots, \theta(T=1-P_{out})]^T, \qquad (6)$$

where $T = 0$ denotes the time of the current computation step. The output weight matrix $\mathbb{W}_{out}$ is the only quantity to be trained in the RC framework to tune the output towards the target signal via a ridge regressions[14] for each computation cycle.

As compared to conventional time-multiplexed physical RCs where $P_{in} = P_{out}$, we explicitly allow these two parameters to be independent and freely adjustable (see Fig. 3b). In particular, we set $P_{out} \gg P_{in}$ for our RC. By this means, each output is derived not only from the reservoir states of its corresponding step, but previous $n_{hist} = P_{out}/P_{in}$ steps. It will be demonstrated in the next sections that this setting enables us to carry out the RC with a good stability of the output and, most importantly, an effective reduction of the impact of the intrinsic noise.

This single physical node architecture can be further extended to multiple physical nodes operated in parallel. In our experiment, we control $N_{node} = 10$ independent physical nodes in one sample simultaneously and set $P_{in} = 2$. The virtual nodes of the 10 physical nodes together constitute the reservoir with a size of $N_{node}P_{in} = 20$. Figure 1d and Supplementary Movie 1 show the real time image and video of the sample in the experiment. Exemplary traces of $\theta(t)$ of four physical nodes (blue lines) driven by an external signal $X$ (red line) in the experiment are plotted in Fig. 3c.

The configuration of the RC is optimized in the simulation for the best performance and then applied in the experiment. The input weights are selected from a binary distribution $\mathbb{W}_{in}^i \in \{-2, 2\}$, which

$P_{in}$ elements, which are sequentially input into the node as the perturbation $u(T)$ of the active particle (Fig. 1c). Each output is linearly derived from the $\theta$ states of previous $P_{out}$ time steps using a weight matrix $\mathbb{W}_{out}$ that is trained via ridge regression. Biases $b_{in/out}$ for the input and output layers are not plotted for the sake of simplicity. **c** Examples of $\theta(t)$ traces of four physical nodes (blue lines) driven by an external signal $X(t)$ (red line, see Eq. (7)) measured in the experiment.

brings a better noise resistance of the RC than the multi-value weights according to simulation results. The details of the RC configuration are described in Supplementary Note 2.

## Chaotic series prediction
We test our RC with the free-running prediction of the chaotic Mackey–Glass series (MGS). The MGS is generated by the delay differential equation

$$\frac{dS(n)}{dn} = \alpha \frac{S(n-\tau)}{1 + S(n-\tau)^\beta} - \gamma S(n), \qquad (7)$$

which was introduced to model the complex dynamics of physiological feedback systems[45]. It has been widely used as a benchmark task for series forecasting[46–49]. With parameters $\alpha = 0.2$, $\beta = 10$, $\gamma = 0.1$, and a delay parameter $\tau = 17$, the MGS exhibits a chaotic behavior with the Lyapunov exponent of around 0.006[46]. The performance of the prediction is evaluated by the normalized root-mean-square error (NRMSE, see Supplementary Note 3 for details). Figure 4 shows the results of the MGS predictions by our RC in experiments and simulations.

**Simulated prediction.** The deterministic simulations have been carried out with a very small reservoir with only 20 virtual nodes but using a large $P_{out} = 400$, i.e., $n_{hist} = 200$ historical reservoir states for each output. The simulation result shows a very good prediction of the target MGS up to around 900 steps (corresponding to 5.4 Lyapunov time) with a NRMSE of $6.7 \times 10^{-2}$ (Fig. 4a). Similarly, Supplementary Note 4 also describes the prediction of a three-dimensional chaotic Lorenz series by our RC in a deterministic simulation. These results signify the capability of our architecture for chaotic systems predictions.

**Experimental prediction.** As compared to the simulations, the experimental RC performance is significantly degraded as a result of the Brownian motion of the active particles and the sensitivity of chaotic systems. The noise induced by Brownian motion acts on the particle angular position $\phi(T)$ (Eq. (3)) and propagates to the virtual node states $\theta(T)$ (Eq. (4)). The signal-to-noise ratio (SNR) of the RC can be estimated by comparing the $\phi(T)$ in simulations with and without noise (see Supplementary Note 5) and reveals an extremely low value (SNR = 1.9 (2.78 dB)). Figure 4b depicts the experimental results from 50

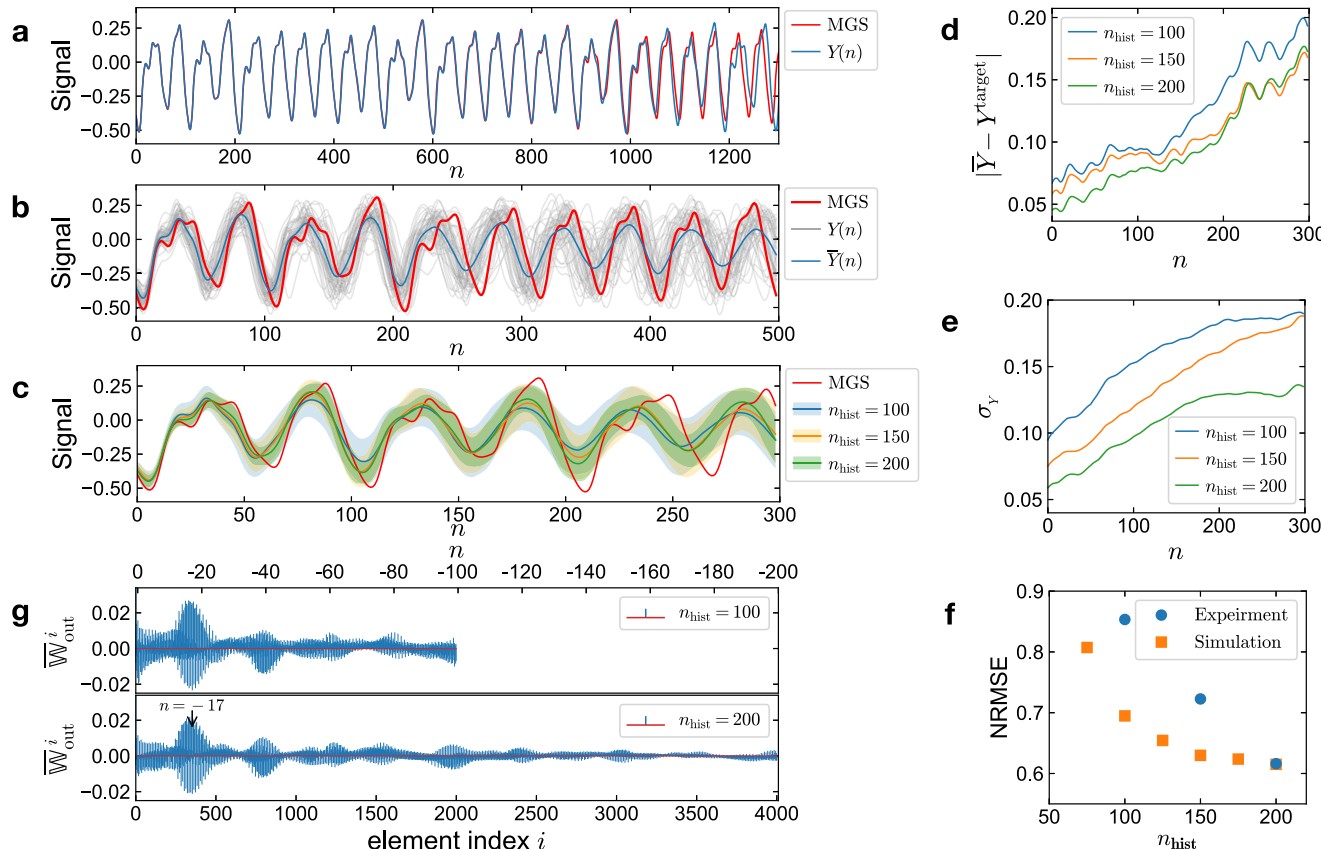

**Fig. 4 | Results of free-running predictions of the Mackey–Glass series.**
**a** Prediction (blue line) of the Mackey–Glass series (MGS, red line) in the determi-nistic simulation by the RC with 20 virtual nodes, and using $n_{hist} = 200$ historical reservoir states ($P_{in} = 2$, $P_{out} = 400$) for each step output. **b** Experimental results of MGS predictions with the same RC configuration as in (**a**). The gray curves denote the RC outputs from 50 repeated predictions with strong fluctuations due to Brownian motion of the active particles. The blue curve represents the mean of the output traces. **c**–**g** Comparison of the experimental results with $P_{out}$ from 200 to 400 ($n_{hist}$ from 100 to 200) evaluated by 50 repeated predictions for each case. **c** Means (colored lines) and corresponding standard deviations (colored areas) of the RC output traces. **d** The deviation between the mean of predictions ($\overline{Y}$) and the target MGS ($Y^{target}$), and **e** the standard deviations $\sigma_Y$ of the predictions versus the step $n$. The curves are smoothed via 100-step moving average. **f** NRMSE of 200 steps predictions from experiments and stochastic simulations versus $n_{hist}$. **g** Means of the output weights $\mathbb{W}_{out}$ trained in experiments. The elements of $\mathbb{W}_{out}$ are in turn the weight for the output bias $b_{out}$, the input signal $X$, and the historical $\theta$ states of the physical nodes (see Supplementary Note 2). Each $N_{node}P_{in} = 20$ weights for $\theta$ correspond to one computation step, which is denoted by $n$ on the top axis with the negative values representing the past. The peak marked by the arrow indicates the high contribution of the historical reservoir states of around $n = -17$, which reveals the property of the target MGS with the delay parameter $\tau = 17$ (Eq. (7)).

repetitions of the MGS prediction. The outputs $Y(n)$ (gray curves) from different experimental predictions exhibit a notable influence of the noise. The mean of the outputs $\overline{Y}(n)$ (blue curve) can reproduce the fundamental period of the target MGS for around 200 steps, and the details of the series for about 50 steps, which is, considering the extremely low SNR, still remarkable. These results obtained from the experimental RC are further underscored by the very good prediction of a non-chaotic periodic trigonometric series (see Supplemen-tary Note 4).

The outstanding RC performance under these very noisy condi-tions becomes possible by introducing the special architecture using a number of historical reservoir states $n_{hist}$ for the output derivation. Figure 4c–g demonstrates the experimental results of the RC with $P_{out}$ varying from 200 to 400, corresponding to $n_{hist}$ from 100 to 200. The accuracy of the predictions is improved by increasing $n_{hist}$ (Fig. 4c). Changing $n_{hist}$ from 100 to 200 results in a decrease of the difference between the mean of the outputs $\overline{Y}$ and the target MGS by 22% (eval-uated by the root-mean-square (RMS) of 200 steps $|\overline{Y} - Y^{target}|$ in Fig. 4d). The standard deviation $\sigma_Y$ of 50 predictions decreases by 32% (200 steps RMS of $\sigma_Y$ in Fig. 4e), and the resulting NRMSE of 200 step predictions diminishes by 28%. This trend of lower prediction error for the experimental system is also picked up by stochastic simulations with an NRMSE reduction of 11% (Fig. 4f).

This finding is striking as increasing $n_{hist}$ does not add more information to the RC, nor increases the reservoir dimensionality, which is determined by the number of linearly independent variables[43,50] of the reservoir. During the computation of the RC, the node states of each step are nonlinearly transformed and mapped into the states of the following steps[43], while they attenuate in magnitude due to the memory fading property of the node. Conventional RCs derive the output from the reservoir state of its corresponding step, which implicitly contains the information of the historical states. Whereas in our RC, the historical reservoir states can directly con-tribute to the output.

The actual contributions of the historical states are determined via the training of the output weights $\mathbb{W}_{out}$. The states correlating more to the current output obtain higher weight magnitudes. Figure 4g plots the $\mathbb{W}_{out}$ trained in experiments with $n_{hist}$ of 100 and 200. The first 2000 elements of $\mathbb{W}_{out}$ corresponding to the near past reservoir states ($n$ from 0 to −99) have similar structures in both cases. As compared to the near past, the far past states ($n$ from −100 to −199 in the $n_{hist} = 200$ case) correspond to smaller weights in amplitude, indicating the weaker correlations and contributions to the current output. The highest magnitude of $\mathbb{W}_{out}$ appears at around $n = -17$, which coincides with the delay parameter $\tau = 17$ of the target MGS (Eq. (7)). The trained $\mathbb{W}_{out}$ thereby partly reveals the property of the target signal.

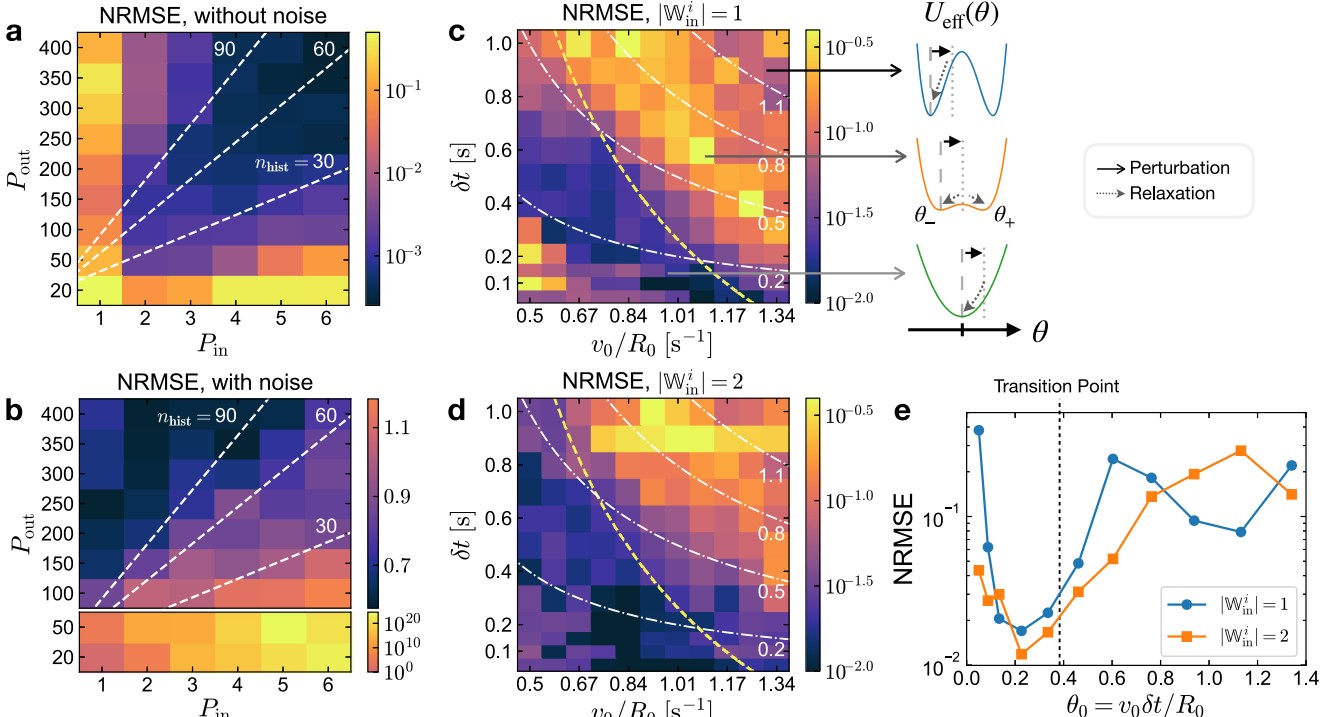

**Fig. 5 | Simulation results of the RC performance as function of the RC configuration.** NRMSE of 200 steps MGS prediction versus $P_{in}$ and $P_{out}$ (**a**, **b**) $v_0/R_0$ and $\delta t$ (**c**–**e**). $\mathbb{W}_{in}$ and $b_{in}$ are optimized (see Supplementary Note 2) for each grid point. **a** Results of RC without noise. The white dashed lines represent the contour lines of $n_{hist} = P_{out}/P_{in}$. **b** Results of RC with noise and the same parameters as in (**a**). Each grid point is evaluated by 50 repeated predictions. The outputs of RC with $P_{out} < 100$ are unstable, and result in large NRMSE as plotted separately in the bottom subplot. **c**, **d** Performance of RC without noise. The elements of $\mathbb{W}_{in}$ are selected from $\{-1, 1\}$ (**c**) and $\{-2, 2\}$ (**d**) respectively. The results with noise are presented in Supplementary Note 6. The white dash-dotted curves denote the contour lines of $\theta_0 = v_0\delta t/R_0$. The yellow dashed curves denote the transition points of $U_{eff}(\theta)$ from single well to double well form. An instrumental feedback latency $\delta t_F = 0.125$ s is considered, thereby the transition curves do not coincide with the $\theta_0$ contour lines. For more details see Supplementary Note 1 and 7. The subplot on the right illustrates the corresponding $U_{eff}(\theta)$, also $\theta$ perturbations and relaxations. **e** Comparison of the NRMSE results of the diagonal grid points in (**c**, **d**) (from lower left to upper right) as function of $\theta_0$. The dashed line denotes the transition point of $U_{eff}$.

**Impact of noise.** These results suggest that there might be an optimal relation between the used number of historical states and the error of the RC to stabilize the prediction and to reduce the impact of the inherent noise due to Brownian motion. To investigate this interrelation we refer to deterministic and stochastic simulations. Figure 5a, b display the RC performance as measured by the NRMSE of the predictions as functions of $P_{in}$ and $P_{out}$. Without noise (Fig. 5a), the results indicate a larger error with low $P_{in}$ or $P_{out}$ and poor performances for $P_{in} < 2$ and $P_{out} < 50$. The performance is improving for increasing $P_{in}$ and $P_{out}$ simultaneously. The best results for the deterministic system are obtained with $n_{hist} = P_{out}/P_{in} = 60$.

For the noisy active particle nodes (Fig. 5b), the RC outputs are unstable for $P_{out} < 100$. The free-running prediction has a high probability to yield fast diverging outputs and a large NRMSE (bottom subplot). With $P_{out} \geq 100$, stable outputs can be obtained. A trend towards lower NRMSE with higher $n_{hist}$ can be observed in agreement with our experimental results (Fig. 4f). Thus, both experiments and simulations confirm that using historical states for the prediction of our RC improves the stability and the quality of the prediction even under extremely noisy conditions.

**Impact of the dynamical properties of the nodes.** Our active particle recurrent node also provides the opportunity to tune the dynamical response of the physical node across the transition of the pitchfork bifurcation with either particle speed $v_0$ or delay $\delta t$. The tuning varies how fast the memory of each node fades, as indicated in Fig. 2d by the relaxation time $\tau_\theta$ and also the coupling of the virtual nodes, as given by Eq. (4).

The impact of the node dynamics on the performance of the RC is depicted in Fig. 5c–e with the NRMSE of the MGS predictions as function of $v_0/R_0$ and $\delta t$ resulting from deterministic simulations. For stochastic simulation results see Supplementary Note 6.

As discussed above, $\theta_0$ determines the asymptotic behavior of $\theta$ after an impulsive perturbation. The correlation between NRMSE and $\theta_0$ with the optimum at around $\theta_0 \approx 0.2$ in Fig. 5e indicates that the dynamical properties of the physical node as characterized by the relaxation time $\tau_\theta$ (Fig. 2d), are indeed a key factor of the RC performance for this task. The optimal performance of the RC is found below the transition point (yellow dashed line) where the approximate effective potential $U_{eff}(\theta)$ is largely determined by its parabolic term (Fig. 5c, right)[27]. At the transition point, where the relaxation time $\tau_\theta$ is largest and the memory fades slowest, the RC does not exhibit the worst performance as expected[9]. The maximal NRMSE appears beyond the transition point, where the $U_{eff}$ is in a double well form. The double well $U_{eff}$ may cause inconsistent responses of the RC, i.e., similar inputs into the node may result in $\theta$ relaxations towards different potential minima. This inconsistency is presumed to induce the degradation of the RC performance[37,51,52]. If the barrier of the double-well potential is further raised by increasing $\theta_0$, the NRMSE appears to decrease again (Fig. 5e) presumably due to the fact that the $\theta$ is largely residing in one of the potential wells with a short relaxation time. This interpretation is supported by the comparison of the RC performances with different $|\mathbb{W}_{in}^i|$ magnitudes in Fig. 5e. Larger $|\mathbb{W}_{in}^i|$ induces stronger inputs as perturbations on $\theta$, thereby a higher probability of $\theta$ to jump over the barrier to another potential well. Hence the maximum of NRMSE with larger $|\mathbb{W}_{in}^i|$ appears at higher $\theta_0$, where the barrier of the potential is also higher.

## Discussion

We have demonstrated above the realization of a physical reservoir computer in an experiment using self-propelled active microparticles. A retarded propulsion towards an immobilized target particle creates a self-organized non-linear dynamical system that is, despite the strong Brownian noise, capable of predicting chaotic time series such as Mackey–Glass and Lorenz series when used as a physical node in a reservoir computer. The key element of this physical recurrent node is a time delay realizing a retarded interaction[27,53,54] that creates the fading memory as a basic requirement for reservoir computing[8]. It also provides the coupling of the active particle dynamics to its past allowing to implement virtual nodes living on a single physical active particle system via time-multiplexing[12,44]. The information processing that is provided by such a single node may therefore extend the rare simulation work on reservoir computing with active particle swarms with interparticle coupling[37]. Additionally, the nonlinearity that is required for computations is an intrinsic physical property of our active particle system and requires no extra treatment of the output signal [55]. Future work could introduce direct physical coupling between the isolated physical nodes in our configuration, e.g., through hydrodynamic or other interactions, to obtain more complex dynamical networks of interacting synthetic active particles.

The dynamics of the physical recurrent node that is the basic unit of our reservoir computer is controlled by a parameter containing the product of activity (active particle speed) and time delay. It can be understood with a simple Landau-like self-induced quartic effective potential, which exhibits a transition from a single-well to double-well form[27]. The system thereby allows to address the relation of RC performance and node relaxation time, where we found a clear indication for an optimal performance below the transition point. While the memory fading becomes extremely slow at the transition point and one might expect the worst performance of the system accordingly, it is observed that a double-well potential with a small barrier leading to inconsistencies in the relaxation dynamics is more severe. Interestingly, such double-well potentials have recently been discussed as nonlinear stochastic-resonance-based activation functions in an attempt to provide better stability of Echo-State-Networks against noise[56].

Noise is an inherent property of our information processing units, as Brownian motion causes strong fluctuations of the node state. Such noises are inevitable at the smallest scales also in the context of biological information processing[57], for instance in neurons[58], with both positive and negative effects[59,60]. In physical RC approaches, noise is commonly a major limiting factor for the performances[43,47,61] although subtle noises are reported to be beneficial as well[8,19,46,62,63]. Yet, general strategies for the noise suppression are unclear except increasing the reservoir size[64], which is normally costly for physical RCs. While the performance of our reservoir computer for the chaotic system prediction is highly degraded by the noise due to the sensitivity of chaotic systems (for non-chaotic series predictions see Supplementary Note 4), we have introduced the architecture that utilizes historical reservoir states for output derivation providing remarkable stability and noise reduction even under low signal-to-noise ratios. This architecture is not increasing the dimensionality of the reservoir nor changing the dynamics of the nodes, and could be potentially useful in future reservoir computing studies.

In summary, simple retarded interactions in synthetic active microparticle systems can give rise to nonlinear self-driven dynamics that form a basis for information processing with active matter. Our reservoir computer highlights this connection between information processing, machine learning, and active matter on the microscale, and also paves the way for new studies on noise in reservoir computing. While we so far referred to isolated active recurrent units, we envision that the high level control of the synthetic active matter will yield new emergent physical collective states, which may leverage the field of active synthetic dynamical systems for information processing.

## Methods

### Sample preparation

The sample used in experiments contains two kinds of microparticles: polystyrene (PS) particles (microParticles GmbH) of 3 μm diameter and the melamine formaldehyde (MF) (microParticles GmbH) particles of 2.19 μm diameter, suspended in a water solution (Fig. 1b). Gold (Au) nano-particles of around 8 nm diameter are uniformly distributed on the surface of the MF particle, cover about 10% of the total surface area of the latter. Two glass coverslips ($20 \times 20$ mm$^2$ and $24 \times 24$ mm$^2$) confine a 3 μm thick sample layer in between. Due to the surface tension of water, the PS particles are compressed and immobilized on the coverslips serving as spacers to define the sample thickness. The PS and MF–Au particles are separately added into two 2% Pluronic F-127 solutions. After 30 min, the Pluronic concentration of both solutions is decreased to 0.02% by diluting twice. Each dilution is followed by a centrifugation and then a removal of a part of the solution to keep the particle concentration. A 0.3 μL sample of PS particles is pipetted on one of the coverslips, then a 0.3 μL sample of MF–Au particles is pipetted in the droplet of PS particles. The sample is then covered carefully with a second coverslip. The edges of the sample are sealed by polydimethylsiloxane (PDMS) to prevent leakage and evaporation, also to relieve the liquid flow inside the sample. The experiment starts about 1 h after the sample preparation to wait until residual liquid flows in the sample ceased.

### Experimental setup

The experimental setup is illustrated in Fig. 1a. The MF–Au microparticles are heated by a focused, continuous-wave laser with a wavelength of 532 nm. The light from the laser module (CNI, MGL-H-532-1W) is expanded by two tube lenses (35 mm, 150 mm focal lengths) in the beam size, then guided by mirrors to a high-speed reflective Spacial Light Modulator (SLM, Meadowlark Optics, HSP512-532), which modulates the phase of the reflected laser. The reflected laser is then guided through two lenses (500 mm, 300 mm focal lengths) to an inverted microscope (Olympus, IX73). A small opaque dot on a glass window located at the focal point of the 500 mm lens serves as a mask to block the unmodulated laser reflected by the SLM top surface. The laser in the microscope is reflected by a dichroic beam splitter (Omega Optical, 560DRLP), then focused by an objective lens (100x, Olympus, UPlanFL N x 100/1.30, Oil, Iris, NA. 0.6–1.3) on the sample plane with a beam width at half maximum about 0.6 μm.

The sample is illuminated by white light from an LED lamp (Thorlabs, SOLIS-3C) through an oil-immersion dark-field condenser (Olympus, U-DCW, NA 1.2–1.4). The image of the sample is projected by the objective lens and a tube lens (180 mm focal length) inside the microscopy stand as well as two additional lenses (100 mm, 150 mm focal lengths) outside the microscopy stand to a camera (Hamamatsu digital sCMOS, C11440-22CU). The numerical aperture (NA) of the objective is set to a value below the minimal NA of the dark-field condenser. Two filters (EKSMA Optics 246-2506-532, Thorlabs FESH0800) in front of the camera block the back reflections of the laser from the microscope. A desktop PC (Intel(R) Core™ i7-7700K CPU @4 × 4.20 GHz, NVIDIA GeForce GTX 1050Ti) with a LabVIEW program (v. 2019) analyzes the images, records data, and manipulates the active particles by controlling the laser through the phase pattern on the SLM. More details are given in Supplementary Note 7.

## Data availability

All data in support of this work is available in the manuscript or the supplementary materials. Further data and materials are available from the corresponding author upon request.

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

## Acknowledgements

This work is supported by Center for Scalable Data Analytics and Artificial Intelligence (ScaDS.AI) Dresden/Leipzig. The authors also acknowledge careful proof reading of the manuscript by Andrea Kramer.

## Author contributions

X.W. and F.C. conceived the experiments. X.W. carried out the experiments and simulations. X.W. and F.C. discussed the results and wrote the manuscript.

## Funding

## Competing interests

The authors declare no competing interests.
