## [Peer Review File · Nature Communications]

Harnessing Synthetic Active Particles for Physical Reservoir ComputingREVIEWER COMMENTS

Reviewer #1 (Remarks to the Author):

The authors present a novel method of physical reservoir computing using synthetic active microparticles. This system stems from a combination of active and passive components, culminating in inherently noisy nonlinear dynamical units. Despite the inherent noise resulting from Brownian motion, the reservoir performs predictive tasks, with noise suppression facilitated by a unique architectural feature that relies on historical reservoir states for output.

Overall this is an interesting work that merges the fields of active matter and reservoir computing, both growing very fast but so far separated. I think it can open a new area of research of fundamental interest and therefore I'd see it well in Nature Communications.

My only major comment is that the practical aspects need to be discussed more in detail. Can this method of reservoir computing be useful for something concrete? or is it only a proof of principle? Either way, I still find the article of seminal importance.

The article is well written and illustrated. So I don't have any major comments on the presentation. As a minor comment: I found a few typos, for example in figure 1, "direction set *by* the vector".

Reviewer #2 (Remarks to the Author):

The authors propose a novel experimental realization of a physical reservoir computer (RC) based on synthetic active particles. The authors combine their experimental results with simulations to provide new insights into the robustness of reservoir computing prediction in the presence of noise in the reservoir. Overall, the paper is well-written and easy to follow. The authors additionally provide detailed sweeps of the free parameters in their system, providing additional insight into the memory decay and the predictivity of the RC. While single-node time delay reservoir computing is not a new concept, the new inclusion of "historical" states in the network architecture, which the authors show significantly improves the robustness of the RC prediction, would be of broad interest to the active matter community, people working physical realizations of RCs, and people studying robust information processing in stochastic environments.

The experimental prediction matches the desired output signal for the non-chaotic trigonometric time series shown in the SI. However, the prediction accuracy degrades significantly for the two chaotic examples in the presence of noise; the predictions, at best, match global trends rather than exact dynamics, especially for the Lorenz examples. I commend the authors for studying chaotic dynamics in the presence of such noise. Simulations of the RC prediction are only shown at a fixed experimentally determined noise level. Could the authors use the computational model at lower noise levels to understand how much noise robustness the inclusion of historical states provides and relate this to the system's Lyapunov exponent? Such an analysis would give helpful information about when the RC architecture proposed is valid.

Furthermore, how does the prediction quality depend on the number of physical nodes - if only a single node is used rather than ten, but the number of independent runs is increased

by ten, does prediction quality decrease? Due to the nonlinearity in the dynamics, I imagine that contemporaneous averaging (increasing the number of physical nodes) does not commute with averaging repeated predictions, perhaps suggesting a relationship between the complexity of the dynamics and the number of physical nodes required.

Minor comments:

Page 2: are intricate examples of this importance and created by evolutionary development (delete and).

Fig 1. caption: direction set "by" the vector from the laser to the particle center (insert by)

Page 8: The signal-to-noise ratio (SNR) of the RC can "be" estimated by (insert be)

Reviewer #3 (Remarks to the Author):

The authors present a demonstration of reservoir computing using the dynamics of an active microparticle as the computational substrate for chaotic time series prediction. To my knowledge it is the first example of a physically realised particle-based reservoir computer (rather than a purely computational implementation). The system is heavily affected by noise and as a result the prediction error is not very remarkable when compared to other RC methods. However, I still found it interesting as a demonstration of a new RC substrate. There are some issues and questions I would like the authors to address before the paper is reconsidered for publication.

What is the advantage of P_{out} being significantly larger than P_{in} . It seems to me that $n_{hist}=100\sim 200$ is very large and there are likely to be a lot of redundant information in those outputs. You mention that this reduces the impact of intrinsic noise, but could this not be equivalently achieved by making the same number of observations with more virtual nodes, possibly with adjustments to the sampling frequency of the MG system? Do you have results with a greater proportion of virtual nodes?

Referring to Figure 5E, it is not clear that there is an optimum at $\theta_0 = 0.2$, or if these are just fluctuations in the graph. How much variability is there in the NRMSE for repetitions of this experiment? Instead it seems that the message from this plot is simply that error increases with increasing θ_0 regardless of the timescale τ_0 . Have you repeated this experiment for the reservoir with noise?

You mention that increasing P_{out} does not increase the reservoir dimensionality, but I think this is missing the fact that the observation of previous states of the system can be reformulated in terms of adding a delay line of recurrent linear connections to the existing structure. This essentially adds a time delay embedding on top of what is already performed by each single node. From this perspective the dimension of the reservoir state will increase.

Reviewer #1 (Remarks to the Author):

The authors present a novel method of physical reservoir computing using synthetic active microparticles. This system stems from a combination of active and passive components, culminating in inherently noisy nonlinear dynamical units. Despite the inherent noise resulting from Brownian motion, the reservoir performs predictive tasks, with noise suppression facilitated by a unique architectural feature that relies on historical reservoir states for output.

Overall, this is an interesting work that merges the fields of active matter and reservoir computing, both growing very fast but so far separated. I think it can open a new area of research of fundamental interest and therefore I'd see it well in Nature Communications.

My only major comment is that the practical aspects need to be discussed in more detail. Can this method of reservoir computing be useful for something concrete? or is it only a proof of principle? Either way, I still find the article of seminal importance.

We thank the reviewer for the comments and questions! The type of physical reservoir computing presented here is a step to understand how dynamical systems formed by synthetic active matter can be employed to process information. This is particularly relevant as these systems on the microscale exhibit strong noise and it is unclear how from this noisy but persistent dynamics information processing might become possible. Though our system is so far still to some extent computer controlled it shows some important ingredients, e.g., the coupling to historical states, how this goal can be achieved. We are working on altering the system to be able to respond to physical processes and have simple applications of reservoir computers controlling the dynamics of active particles already in place.

The article is well written and illustrated. So I don't have any major comments on the presentation. As a minor comment: I found a few typos, for example in figure 1, "direction set *by* the vector".

Thank you very much for spotting these typos! We have carefully revised the manuscript and hope that we have eliminated all the residual typos!

Reviewer #2 (Remarks to the Author):

The authors propose a novel experimental realization of a physical reservoir computer (RC) based on synthetic active particles. The authors combine their experimental results with simulations to provide new insights into the robustness of reservoir computing prediction in the presence of noise in the reservoir. Overall, the paper is well-written and easy to follow. The authors additionally provide detailed sweeps of the free parameters in their system, providing additional insight into memory decay and the predictivity of the RC. While single-node time delay reservoir computing is not a new concept, the new inclusion of "historical" states in the network architecture, which the authors show significantly improves the robustness of the RC prediction, would be of broad interest to the active matter community, people working physical realizations of RCs, and people studying robust information processing in stochastic environments.

The experimental prediction matches the desired output signal for the non-chaotic trigonometric time series shown in the SI. However, the prediction accuracy degrades significantly for the two chaotic examples in the presence of noise; the predictions, at best, match global trends rather than exact dynamics, especially for the Lorenz examples. I commend the authors for studying chaotic dynamics in the presence of such noise. Simulations of the RC prediction are only shown at a fixed experimentally determined noise level. **Could the authors use the computational model at lower noise levels to understand how much noise robustness the inclusion of historical states provides and relate this to the system's Lyapunov exponent? Such an analysis would give helpful information about when the RC architecture proposed is valid.**

Thank you very much for your comments and suggestions! They help us to improve our manuscript. The plot below shows the RC performance (NRMSE) as a function of the noise level, which is represented by the diffusivity of the Brownian motion D (D_0 is that in experiment), with different number of historical states n_{hist} for output derivation.

To test the robustness of our RC system, we add a perturbation ε into the input of the RC and compare the outputs with and without perturbation (O^* and O) in deterministic simulation. The relative output deviation is plotted below.

Due to the oscillatory property of the Mackey-Glass series, it is difficult to determine the Lyapunov exponent. Instead, we define the parameter Λ , where the relative deviation of the outputs exceeds e . Larger n_{hist} , i.e., more historical states involved, more robust the system is. Detailed descriptions are added in Sec. 8 of Supplementary Information now.

Furthermore, how does the prediction quality depend on the number of physical nodes - if only a single node is used rather than ten, but the number of independent runs is increased by ten, does prediction quality decrease? Due to the nonlinearity in the dynamics, I imagine that contemporaneous averaging (increasing the number of physical nodes) does not commute with averaging repeated predictions, perhaps suggesting a relationship between the complexity of the dynamics and the number of physical nodes required.

The plots below show the prediction performance (NRMSE) as a function of the number of physical nodes with and without noise. To have a fair comparison, the P_{in} for each RC is adjusted to keep their number of the virtual nodes (i.e., number of physical nodes $\times P_{\text{in}}$) the same as 20.

Therefore, larger number of physical nodes corresponds to smaller P_{in} , which means sparser coupling between the virtual nodes (because different physical nodes have no coupling between).

For the deterministic case, a trend towards lower NRMSE with larger number of physical nodes can be observed in the results. This trend agrees with the fact that the sparser nodes interconnection normally leads to better RC performance [1]. For the RC with noise, the NRMSE does not show a significant dependency on the number of the physical nodes.

1.

[1] Lukoševičius M. A Practical Guide to Applying Echo State Networks. *Neural Networks: Tricks of the Trade*. Vol 7700. Lecture Notes in Computer Science. Springer Berlin Heidelberg; 2012:659-686.

Minor comments:

Page 2: are intricate examples of this importance and created by evolutionary development (delete and).

Fig 1. caption: direction set "by" the vector from the laser to the particle center (insert by)

Page 8: The signal-to-noise ratio (SNR) of the RC can "be" estimated by (insert be)

Thank you very much! We have corrected the typos and revised the manuscript.

Reviewer #3 (Remarks to the Author):

Thank you very much for the excellent questions and comments, which helped us to improve the manuscript!

The authors present a demonstration of reservoir computing using the dynamics of an active microparticle as the computational substrate for chaotic time series prediction. To my knowledge it is the first example of a physically realised particle-based reservoir computer (rather than a purely computational implementation). The system is heavily affected by noise and as a result the prediction error is not very remarkable when compared to other RC methods. However, I still found it interesting as a demonstration of a new RC substrate. There are some issues and questions I would like the authors to address before the paper is reconsidered for publication.

What is the advantage of P_{out} being significantly larger than P_{in} . It seems to me that $n_{hist}=100\sim 200$ is very large and there are likely to be a lot of redundant information in those outputs.

Yes. The number of historical states used to generate the output can be very large. If the information is relevant to the output is decided by the training. The resulting output weights W_{out} (Fig. 4G) will typically exclude highly redundant information. This can be seen, for example, in Fig. 4G, where the virtual node states from the far past obtain only small W_{out} weights. In general, the value of used historical states, as we think, is not very large given the fact that the reservoir is extremely small.

Do you have results with a greater proportion of virtual nodes?

Yes. Fig. 5 A, B show these simulation results. The total number of virtual nodes is the number of physical nodes $\times P_{in}$. Fig. 5 A, B demonstrate the reservoir computing with P_{in} from 1 to 6, corresponding to 10 to 60 virtual nodes.

You mention that this reduces the impact of intrinsic noise, but could this not be equivalently achieved by making the same number of observations with more virtual nodes, possibly with adjustments to the sampling frequency of the MG system?

Thank you very much for this interesting question. According to other studies (e.g. reference [2], see below), increasing the number of virtual nodes can indeed reduce the noise.

But we think it is difficult to judge whether increasing the number of real/virtual nodes and increasing n_{hist} are equivalent. In the first method, node states are used in a nonlinear fashion via the connection between the virtual nodes, whereas in the second, they are used in a linear way.

From a practical perspective, increasing the number of (virtual) nodes is normally difficult and costly for physical RC in experiment. While increasing n_{hist} is much easier and only refers to linear data processing and does not conflict the former method. The latter hence has a big advantage in real physical RC applications. Nevertheless, this interesting topic is worth further theoretical investigation in future, also taking into consideration the computational effort that is inherent to both ways.

[2] Alata, et. al. 2020. "Phase Noise Robustness of a Coherent Spatially Parallel Optical Reservoir." *IEEE Journal of Selected Topics in Quantum Electronics* 26

Referring to Figure 5E, it is not clear that there is an optimum at $\theta_0 = 0.2$, or if these are just fluctuations in the graph. How much variability is there in the NRMSE for repetitions of this experiment? Instead, it seems that the message from this plot is simply that error increases with increasing θ_0 regardless of the timescale τ_0 .

Thank you very much for pointing this out! We updated Fig. 5E, as plotted below.

The optimum and the correlation between NRMSE and θ_0 can be observed in color plots Fig. 5C, D. Indeed, they were not obvious in Fig. 5E, which was showing the data columns of Fig. 5C, D ($v_0/R_0 = 1.01$ column). The updated figure now plots the diagonal grid points of Fig. 5C and D (from lower left to upper right). The NRMSE is now showing the minimum at $\theta_0 \sim 0.2$, and maxima dependent on $|W_{in}|$.

Have you repeated this experiment for the reservoir with noise?

The NRMSE vs. θ_0 relation with noise is investigated in the stochastic simulation. The results are shown below, and detailedly discussed in Sec. S3 of Supplementary Information. They indicate about one order of magnitude larger NRMSE than the deterministic case, but similar correlation of NRMSE vs. θ_0 .

You mention that increasing P_{out} does not increase the reservoir dimensionality, but I think this is missing the fact that the observation of previous states of the system can be reformulated in terms of adding a delay line of recurrent linear connections to the existing structure. This essentially adds a time delay embedding on top of what is already performed by each single node. From this perspective the dimension of the reservoir state will increase.

The W_{out} and P_{out} are only computational technics for extracting information from the reservoir. These parameters do not influence the dynamics of the physical nodes (plot A below). The dynamics is only affected by the input signals and the intrinsic properties of the active particle (namely v_0 and delay, as Eq. 3 and 4 in the manuscript). From the perspective of the RC, the input is projected to the high dimension space constructed by the node dynamics. Since the dynamics is unrelated to P_{out} or W_{out} , we claim that the dimensionality of RC is not changed by P_{out} .

From the perspective of a recurrent neuronal network, the Reservoir Computer can be deemed as a feedforward neuronal network by unfolding along time (plot B below). This structure is, as you have stated, equivalent to a network with an expanded number of neurons per layer and additional linear connection (more precisely, identity connection) between them (plot C). The number of virtual nodes per layer is displayed as P_{out} . However, only P_{in} virtual nodes in the layer have non-linear connection (black arrows in plot C) in this topology. The rest $P_{out} - P_{in}$ virtual nodes are just linearly connected to the virtual nodes of the last layer and contain no additional degree of freedom or new information. Since the RC dimensionality is determined by linearly independent neurons [3,4], we believe that increasing P_{out} does not change the RC dimensionality.

In our work, we propose for the first time an architecture that utilizes historical node states, so theoretical investigations on the dimensionality of this particular case are still lacking. This topic is worth further investigation in the future.

[3] Dambre, J.; Verstraeten, D.; Schrauwen, B.; Massar, S. Information Processing Capacity of Dynamical Systems. *Sci Rep* **2012**, *2* (1), 514.

[4] Carroll, T. L. Dimension of Reservoir Computers. *Chaos: An Interdisciplinary Journal of Nonlinear Science* **2020**, *30* (1), 013102.

REVIEWERS' COMMENTS

Reviewer #1 (Remarks to the Author):

I believe the authors have fully addressed the remaining points and the article can now be published.

Reviewer #2 (Remarks to the Author):

The authors have carefully addressed my questions in their revised version, and I am strongly recommending this very interesting work for publication in Nature Communications.

Reviewer #3 (Remarks to the Author):

I thank the authors for their responses to my questions, which have clarified a lot. However, I disagree with their comments regarding the the effect of P_{out} on the dimension of the reservoir. Even though the dimension of the nonlinear node itself is unaffected by observing the history of states, I think it is incorrect to say that the dimension of the reservoir itself is unchanged. Of course, there is nothing wrong with this approach, as the use of historical states is a very efficient way to increase the dimension of the reservoir and it is a way to improve reservoir performance without increasing the amount of costly nonlinear processing. But, I think that statements about increased performance without increased dimensionality are misleading and I would like the authors to reconsider this point. Overall, I am happy with the authors responses and the changes that they have made and would recommend the paper for publication with optional changes on this point regarding dimension.